**Data Availability Statement:** All data underlying the results presented in this study were collected from Kilimanjaro Christian Medical Centre (KCMC)

# An analysis of emergency care delays experienced by traumatic brain injury patients presenting to a regional referral hospital in a low-income country

Armand Zimmerman[1], Samara Fox[2], Randi Griffin[3], Taylor Nelp[4], Erika Bárbara Abreu Fonseca Thomaz[5], Mark Mvungi[6], Blandina T. Mmbaga[6,7,8], Francis Sakita[6,8], Charles J. Gerardo[4], Joao Ricardo Nickenig Vissoci[1,4], Catherine A. Staton[1,4]*

1 Duke Global Health Institute, Duke University, Durham, North Carolina, United States of America, 2 Yale School of Medicine, New Haven, Connecticut, United States of America, 3 Department of Evolutionary Anthropology, Duke University, Durham, North Carolina, United States of America, 4 Division of Emergency Medicine, Department of Surgery, Duke University Medical Center, Durham, North Carolina, United States of America, 5 Federal University of Maranhão, Maranhão, Brazil, 6 Kilimanjaro Christian Medical Centre, Moshi, Tanzania, 7 Kilimanjaro Clinical Research Institute, Moshi, Tanzania, 8 Kilimanjaro Christian Medical University College, Moshi, Tanzania

* catherine.staton@duke.edu

## Abstract

### Background

Trauma is a leading cause of death and disability worldwide. In low- and middle-income countries (LMICs), trauma patients have a higher risk of experiencing delays to care due to limited hospital resources and difficulties in reaching a health facility. Reducing delays to care is an effective method for improving trauma outcomes. However, few studies have investigated the variety of care delays experienced by trauma patients in LMICs. The objective of this study was to describe the prevalence of pre- and in-hospital delays to care, and their association with poor outcomes among trauma patients in a low-income setting.

### Methods

We used a prospective traumatic brain injury (TBI) registry from Kilimanjaro Christian Medical Center in Moshi, Tanzania to model nine unique delays to care. Multiple regression was used to identify delays significantly associated with poor in-hospital outcomes.

### Results

Our analysis included 3209 TBI patients. The most common delay from injury occurrence to hospital arrival was 1.1 to 4.0 hours (31.9%). Most patients were evaluated by a physician within 15.0 minutes of arrival (69.2%). Nearly all severely injured patients needed and did not receive a brain computed tomography scan (95.0%). A majority of severely injured patients needed and did not receive oxygen (80.8%). Predictors of a poor outcome included delays to lab tests, fluids, oxygen, and non-TBI surgery.

in Moshi, Tanzania with permission from the KCMC Ethics Committee. It is KCMC institutional practice to mediate data access through individuals rather than sharing openly. The data used for this study are available from Gwamaka William (email: gwamakawilliam14@gmail.com). Gwamaka William is a non-author, KCMC institutional point of contact who is able to field data access queries for this publication.

**Funding:** This project was made possible by the Mentored Research Training Program in collaboration with the HRSA-funded KCMC MEPI grant # T84HA21123-02; U.S. National Institutes of Health and the Duke Division of Emergency Medicine. Randi Griffin would like to acknowledge support from the NSF Graduate Research Fellowship. Dr. Staton would like to acknowledge salary support funding from the Fogarty International Center (Staton, K01 TW010000-01A1). The funders had no role in study design, data collection and analysis, decision to publish, or preparation of the manuscript.

**Competing interests:** The authors have declared that no competing interests exist.

## Conclusions

Time to care data is informative, easy to collect, and available in any setting. Our time to care data revealed significant constraints to non-personnel related hospital resources. Severely injured patients with the greatest need for care lacked access to medical imaging, oxygen, and surgery. Insights from our study and future studies will help optimize resource allocation in low-income hospitals thereby reducing delays to care and improving trauma outcomes in LMICs.

## Introduction

An estimated 69 million people worldwide experience a traumatic brain injury (TBI) each year [1]. The distribution of this burden is biased towards low- and middle-income countries (LMICs) which endure 90% of global injury related deaths and three times more TBI cases than high-income countries [1–3]. Among LMICs the greatest burden of TBI exists in the World Health Organization (WHO) Africa Region, which accounts for 15.9% and 11.5% of LMIC and global TBIs respectively [1]. Furthermore, predictive models have estimated a future annual TBI incidence of up to 14 million cases in Africa by the year 2050 [4].

The notable presence of TBI in African countries is likely attributable to excessive road traffic incidents precipitated by increased motor vehicle use in combination with poor compliance to road safety practices and lagging infrastructural development [5–7]. This is particularly true in sub-Saharan Africa where the cumulative incidence of TBIs due to road traffic incidents is 156 per 100,000 people; significantly higher than the global value of 106 per 100,000 people [2]. However, in addition to enduring a high TBI incidence, patients who sustain a TBI in sub-Saharan Africa have disproportionately worse outcomes in comparison to patients in high income countries [8–12]. The cause of this disparity may result largely from differences in the quality of prehospital and in-hospital care [8, 13, 14]

One widely recognized measure of hospital care quality, particularly for emergency departments (EDs), is time to care [15–18]. In high-income countries (HICs) delays to care within hospital EDs are frequent and have been shown to result in worse outcomes, including higher mortality rates [19–27]. However, information from LMICs regarding the extent of ED delays and their impact on patient outcomes is sparse [28–34]. Moreover, such data is lacking for TBI patients presenting to EDs in sub-Saharan Africa. There is a need to investigate the relationship between ED care delays and in-hospital outcomes among TBI patients in sub-Saharan Africa if we are to improve trauma outcomes in this region of the world.

Tanzania is a low-income country in East sub-Saharan Africa with a large TBI burden. The TBI prevalence ranges from 21% in urban areas to 34% in rural areas and mortality rates have reached upwards of 30% for patients presenting to large referral hospitals including Muhimbili National Hospital in Dar es Salaam and Bugando Medical Centre in Mwanza [35–37]. Although Tanzania has recently started an emergency medicine residency program and a trauma care quality improvement system, the country does not yet have national guidelines for emergency and critical care [38]. Consequently, many hospitals lack staff trained in the triage and management of critically injured patients [38]. Trauma patients in low-income settings like Tanzania may therefore experience longer delays to care and thus worse outcomes when compared to trauma patients in HICs. The objective of this study was to describe the prevalence of pre- and in-hospital care delays and their association with poor outcomes among trauma patients presenting to an emergency department (ED) in a low-income hospital.

## Methods

### Study design

This study is a secondary analysis of a TBI patient registry established at Kilimanjaro Christian Medical Center (KCMC) in Moshi, Tanzania. The registry prospectively enrolled patients presenting to the KCMC ED for treatment of their acute TBI (<24 hours since injury occurrence) from May 2013 to August 2017 [12]. The registry includes information on demographics, injury status, vital signs, treatment, time of treatment, injury management, and injury outcomes.

### Study setting

KCMC is a tertiary referral hospital in Moshi, Tanzania serving over 15 million people [39]. Nearly 1000 TBI patients present to the ED each year with about 33% requiring admission to the intensive care unit (ICU) [12]. The mortality rate of patients presenting to KCMC with severe TBI is close to 50% [12]. The hospital contains 630 patient beds and employs 1300 staff. However, the ED houses only six adult patient beds and three physicians yet receives an average of 80 to 100 patients a day [39]. TBI surgeries are limited to burr holes, craniotomies, and craniectomies conducted by trained general surgeons, as there are no neurosurgeons stationed at KCMC. The decision to operate on a TBI patient and what kind of operation to perform is ultimately based on the judgement of a general surgeon trained in neurosurgical procedures. Alternatively, a patient or patient's family may choose to forgo surgery if the cost cannot be covered.

### Study participants

Inclusion into the TBI registry was restricted to adult patients (≥18 years) seeking care for acute TBI who survived to evaluation by an ED physician. Patients who presented with follow-up and non-acute injuries or those who died prior to physician evaluation were not eligible for inclusion. All patients who were enrolled in the registry were included in our analysis.

### Study variables

Variables selected for inclusion into this analysis were age, gender, mechanism of injury, alcohol use, respiratory rate, systolic blood pressure, oxygen saturation, heart rate, Glasgow coma score (GCS), transfer status, and time to care. Transfer status was categorized as transferred or not transferred to KCMC from a previous hospital. Patients with missing prehospital course information were categorized as having an unknown transfer status. Time to care variables included time from (1) injury to hospital arrival, (2) hospital arrival to physician arrival, (3) physician arrival to lab tests sent to the laboratory, (4) physician arrival to chest or skull x ray, (5) physician arrival to brain computed tomography (CT) scan, (6) physician arrival to fluids administered, (7) physician arrival to oxygen administered, (8) physician arrival to TBI surgery, and (9) physician arrival to non-TBI surgery. The chronology of these time intervals is depicted in Fig 1. The outcome variable used in this analysis was Glasgow outcome score (GOS) dichotomized as good (4 or 5) and poor (1, 2, or 3). Among patients who survived, GOS was measured at hospital discharge. Hospital discharge was our last point of contact with patients included in our registry. We report the mean time from injury to hospital discharge as well as the mean time from injury to death.

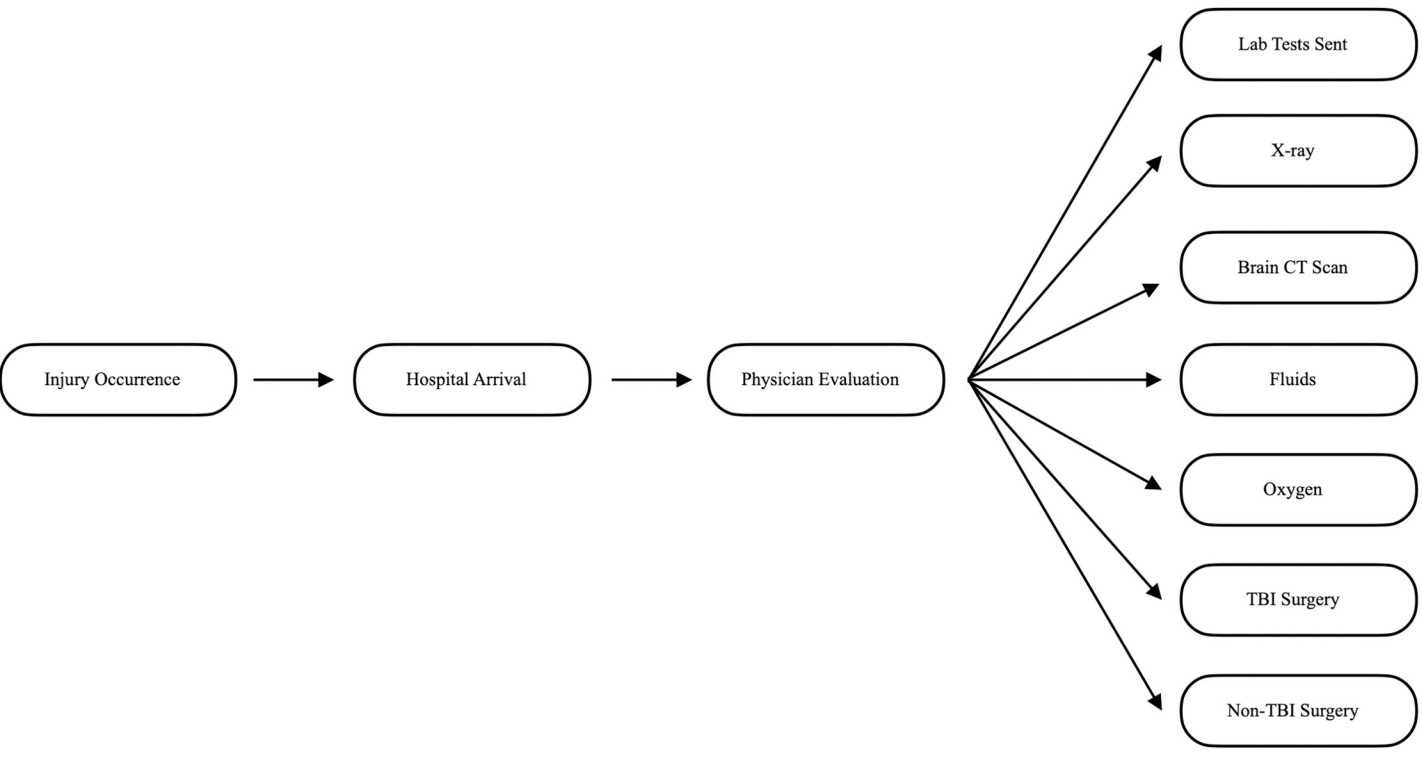

**Fig 1. Chronological sequence of time intervals.** Each arrow represents one of our nine time to care variables.

## Time standards

The South African Triage Scale (SATS) is a protocol that was designed to improve patient triage in resource limited settings around South Africa [40]. Since its development, the SATS has been validated and proven reliable in both urban and rural South African hospitals as well as in numerous LMICs [41–47]. Based on time to care standards outlined in the SATS, we considered the following time intervals to categorize our nine time to care variables: 0.0–1.0 hours, 1.1–4.0 hours, 4.1–12.0 hours, and greater than 12.0 hours.

## Missing time data

Systolic blood pressure and oxygen saturation were used to identify patients with hypotension or hypoxia on arrival. Binary variables were created for hypotension (<100 mmHg systolic blood pressure) and hypoxia (<92% pulse oxygen). Using these binary variables and GCS, we created two additional levels for three of our time to care variables: (1) patient did not receive the procedure and needed the procedure, and (2) patient did not receive the procedure and did not need the procedure. These two categories were used to classify patients when data was missing for time to brain CT scan, time to fluids, and time to oxygen.

We defined needing a brain CT scan as presenting with a GCS less than 13. Needing fluids was defined as being hypotensive. Needing oxygen was defined as having hypoxia or a GCS less than 8. For patients who received oxygen, the availability of time data was insufficient for analysis. We therefore categorized all oxygen recipients as having received oxygen. For time to TBI surgery and time to non-TBI surgery, we had no means by which to identify a patient's need for surgery. As a result, if a patient had missing data for these two variables and if they did not receive surgery, then he or she was classified as having not received surgery.

## Data analysis

**Imputation.** Variables with more than 20% of observations missing were excluded from our final analysis. Two variables were removed during this process: (1) time from physician arrival to mannitol administration, and (2) time from physician arrival to surgeon arrival. For the remaining variables, we used multiple imputation by chained equations to impute missing values. This was achieved through the mice package in R Language for Statistical Computing. All data analysis was performed using R software.

**Descriptive statistics.** We computed descriptive statistics for all variables stratified by our GOS outcome metric. A majority of patients in our registry who experienced a poor GOS died. Consequently, we do not differentiate our results by mortality in addition to differentiating by good and poor GOS. For continuous variables we report the mean and standard deviation. For categorical variables we report frequencies and percentages. For continuous variables, t-tests were used to identify significant differences ($\alpha < 0.05$) in means between the good and poor outcome groups. For categorical variables, chi-square or Fisher's exact tests were used to identify significant differences ($\alpha < 0.05$) in the distribution of observations between the good and poor outcome groups.

**Inferential statistics.** We used logistic regression to model GOS as a function of time to care. We subsequently adjusted our models for age, gender, mechanism of injury, respiratory rate, systolic blood pressure, oxygen saturation, heart rate, and GCS. For all models we report odds ratios and 95% confidence intervals (CIs). For all models, a good outcome (GOS 4–5) was used as the reference level for our outcome variable. For time to care variables, the lowest time interval was used as the reference level. In the case of time to oxygen, there was insufficient time data to form useful time interval levels. As a result, having received oxygen was used as the reference level.

## Ethics

This study was approved by the KCMC Ethics Committee and the National Institute of Medical Research, Tanzania as well as by the Duke University Institutional Review Board. Participant consent was waived as this study is a secondary analysis of a de-identified patient registry.

## Results

### Sociodemographic, injury, and clinical characteristics

Our sample included 3209 patients. Patients were predominantly male (82.2%) and had a mean age of 32.1 years (±16.5). Most patients suffered TBI due to a road traffic incident (67.7%). With regard to GCS, most patients were classified as mild (78.4%) followed by severe (12.5%) and moderate (9.1%). Mean vital signs of our sample are presented in Table 1. Many patients were transferred to KCMC from a neighboring hospital (21.3%). Overall, 2848 (88.8%) and 361 (11.2%) patients experienced a good and poor outcome respectively. Of the 361 patients who experienced a poor outcome, 326 patients (90.3%) died. The mean time from injury to death was 3.6 days (SD: 3.5 days). Among the remaining 2883 patients who survived, the mean time from injury to hospital discharge was 4.8 days (SD: 4.6 days).

### Time to arrival

The most common wait time from injury occurrence to hospital arrival was 1.1 to 4.0 hours (31.9%) followed by 12.1 hours or more (31.7%), 0.0 to 1.0 hours (18.9%), and 4.1 to 12.0 hours (17.5%) (Table 2). Delays from injury occurrence to hospital arrival were not different between mild, moderate, and severe GCS patients ($p > 0.05$) (Fig 2). However, of patients

**Table 1. Demographic and clinical characteristics and crude association with a poor outcome.**

| Variable | Total | Good Outcome | Poor Outcome | p-Value | Odds Ratio (95% CI) | p-Value |
|---|---|---|---|---|---|---|
| **Age, Mean (SD)** | 32.1 (16.5) | 31.5 (16.3) | 36.3 (17.7) | <0.001 | 1.02 (1.01–1.02) | <0.001 |
| **Sex, N (%)** | | | | | | |
| Female | 572 (17.8) | 516 (90.2) | 56 (9.8) | | ref | |
| Male | 2637 (82.2) | 2332 (88.4) | 305 (11.6) | 0.243 | 1.21 (0.90–1.64) | 0.224 |
| **Mechanism of Injury, N (%)** | | | | | | |
| Assault | 462 (14.4) | 434 (93.9) | 28 (6.1) | | ref | |
| Road Traffic Injury | 2171 (67.7) | 1911 (88.0) | 260 (12.0) | | 2.11 (1.43–3.22) | <0.001 |
| Fall | 347 (10.8) | 293 (84.4) | 54 (15.6) | | 2.86 (1.78–4.67) | <0.001 |
| Other | 229 (7.1) | 210 (91.7) | 19 (8.3) | <0.001 | 1.40 (0.76–2.55) | 0.274 |
| **Glasgow Coma Score, N (%)** | | | | | | |
| Mild | 2517 (78.4) | 2438 (96.9) | 79 (3.1) | | ref | |
| Moderate | 291 (9.1) | 225 (77.3) | 66 (22.7) | | 9.05 (6.34–12.90) | <0.001 |
| Severe | 401 (12.5) | 185 (46.1) | 216 (53.9) | <0.001 | 36.03 (26.87–48.78) | <0.001 |
| **Respiratory Rate, Mean (SD)** | 21.9 (5.0) | 21.7 (4.8) | 23.0 (6.5) | 0.020 | 1.04 (1.02–1.05) | <0.001 |
| **Systolic Blood Pressure, Mean (SD)** | 121.7 (21.6) | 121.6 (20.5) | 122.5 (28.6) | <0.001 | 1.00 (1.00–1.01) | 0.478 |
| **Pulse Oxygen, Mean (SD)** | 95.4 (7.5) | 96.2 (5.6) | 88.8 (14.3) | 0.006 | 0.92 (0.91–0.93) | <0.001 |
| **Heart Rate, Mean (SD)** | 88.0 (18.1) | 87.5 (17.1) | 91.9 (24.1) | <0.001 | 1.01 (1.01–1.02) | <0.001 |
| **Transfer Status, N (%)** | | | | | | |
| Not Transferred | 370 (11.5) | 341 (92.2) | 29 (7.8) | | ref | |
| Transferred | 684 (21.3) | 608 (88.9) | 79 (11.1) | | 1.47 (0.95, 2.33) | 0.092 |
| Unknown | 2155 (67.2) | 1899 (88.1) | 256 (11.9) | 0.069 | 1.59 (1.08, 2.41) | 0.024 |

with a known transfer status, most who arrived more than 4.0 hours after injury occurrence were transferred from a neighboring hospital (85.5%) (Fig 3). In addition, those with longer hospital arrival delays tended to be transferred patients (p<0.001). We found no significant associations between time to arrival and in-hospital outcome in our adjusted regression model (Fig 4).

## Time to physician evaluation

Upon hospital arrival, most patients waited 0.0 to 15.0 minutes to be evaluated by a physician in the ED (69.2%), followed by 15.1 to 30.0 minutes (19.0%), more than 45.0 minutes (7.8%), and 30.1 to 45.0 minutes (4.1%) (Table 2). A majority of mild (66.9%), moderate (71.1%), and severe (81.8%) GCS patients waited 15.0 minutes or less to be evaluated by an ED physician (Fig 2). Nearly all patients were evaluated within 45.0 minutes (92.2%). We found no significant associations between time to physician evaluation and in-hospital outcome in our adjusted regression model (Fig 4).

## Time to diagnostics

Following physician evaluation, most patients waited 0.0 to 1.0 hours to receive lab tests (48.4%) (Table 2). A majority of severe GCS patients (56.1%) received lab tests within 1.0 hours of physician evaluation whereas a majority of mild (52.0%) and moderate (53.0%) GCS patients received lab tests more than 1.0 hours after evaluation. In our adjusted regression model, receiving lab tests 1.1 to 4.0 hours after physician evaluation was protective against a poor outcome in comparison to those receiving lab tests within 1.0 hours of evaluation (OR: 0.61; 95% CI: 0.43, 0.86) (Fig 4).

**Table 2. Delays to care and crude association with a poor outcome.**

| Variable | | Total | Good Outcome | Poor Outcome | p-Value | Odds Ratio (95% CI) | p-Value |
|---|---|---|---|---|---|---|---|
| **Time to Arrival, N (%)** | 0.0–1.0 h | 607 (18.9) | 536 (87.6) | 71 (12.4) | | ref | |
| | 1.1–4.0 h | 1023 (31.9) | 921 (88.8) | 102 (11.2) | | 0.84 (0.61–1.16) | 0.274 |
| | 4.1–12.0 h | 562 (17.5) | 497 (89.0) | 65 (11.0) | | 0.99 (0.69–1.41) | 0.944 |
| | >12.0 h | 1017 (31.7) | 894 (89.3) | 123 (10.7) | 0.450 | 1.04 (0.76–1.42) | 0.811 |
| **Time to Physician, N (%)** | 0.0–15.0 m | 2220 (69.2) | 1935 (87.0) | 285 (13.0) | | ref | |
| | 15.1–30.0 m | 609 (19.0) | 553 (91.6) | 56 (8.4) | | 0.69 (0.50–0.92) | 0.015 |
| | 30.1–45.0 m | 132 (4.1) | 123 (93.1) | 9 (6.9) | | 0.50 (0.23–0.93) | 0.046 |
| | >45.0 m | 248 (7.8) | 237 (94.8) | 11 (5.2) | <0.001 | 0.32 (0.16–0.56) | <0.001 |
| **Time to Labs Sent, N (%)** | 0.0–1.0 h | 1554 (48.4) | 1340 (86.1) | 214 (13.9) | | ref | |
| | 1.1–4.0 h | 1111 (34.6) | 1010 (91.5) | 101 (8.5) | | 0.63 (0.49–0.80) | <0.001 |
| | 4.1–12.0 h | 347 (10.8) | 323 (92.6) | 24 (7.4) | | 0.47 (0.29–0.71) | <0.001 |
| | >12.0 h | 197 (6.1) | 175 (87.3) | 22 (12.7) | <0.001 | 0.79 (0.48–1.23) | 0.314 |
| **Time to X-ray, N (%)** | 0.0–1.0 h | 1647 (51.3) | 1427 (86.7) | 220 (13.3) | | ref | |
| | 1.1–4.0 h | 1234 (38.5) | 1123 (90.4) | 111 (9.6) | | 0.64 (0.50–0.81) | <0.001 |
| | >4.0 h | 328 (10.2) | 298 (93.0) | 30 (7.0) | <0.001 | 0.65 (0.43–0.96) | 0.037 |
| **Time to Brain CT, N (%)** | 0.0–1.0 h | 39 (1.2) | 31 (79.5) | 8 (20.5) | | ref | |
| | 1.1–4.0 h | 55 (1.7) | 45 (81.8) | 10 (18.2) | | 0.86 (0.31–2.49) | 0.777 |
| | >4.0 h | 51 (1.6) | 45 (88.2) | 6 (11.8) | | 0.52 (0.16–1.63) | 0.262 |
| | Not Received, Needed | 778 (24.2) | 503 (64.7) | 275 (35.3) | | 2.12 (1.01–5.01) | 0.063 |
| | Not Received, Not Needed | 2286 (71.2) | 2224 (97.3) | 62 (2.7) | <0.001 | 0.12 (0.05–0.26) | <0.001 |
| **Time to Fluids, N (%)** | 0.0–1.0 h | 821 (25.6) | 681 (82.9) | 140 (17.1) | | ref | |
| | 1.1–4.0 h | 105 (3.3) | 90 (85.7) | 15 (14.3) | | 0.81 (0.44–1.40) | 0.475 |
| | >4.0 h | 128 (4.0) | 112 (87.5) | 16 (12.5) | | 0.69 (0.39–1.18) | 0.198 |
| | Not Received, Needed | 150 (4.7) | 127 (84.7) | 23 (15.3) | | 0.88 (0.53–1.40) | 0.605 |
| | Not Received, Not Needed | 2005 (62.5) | 1838 (91.7) | 167 (8.3) | <0.001 | 0.44 (0.35–0.56) | <0.001 |
| **Time to Oxygen, N (%)** | Received | 104 (3.2) | 33 (31.7) | 71 (68.3) | | ref | |
| | Not Received, Needed | 513 (16.0) | 341 (66.5) | 172 (33.5) | | 0.23 (0.15, 0.37) | <0.001 |
| | Not Received, Not Needed | 2592 (80.8) | 2474 (95.4) | 118 (4.6) | <0.001 | 0.02 (0.01, 0.03) | <0.001 |
| **Time to TBI Surgery, N (%)** | 0.0–12.0 h | 304 (9.5) | 266 (87.3) | 38 (12.7) | | ref | |
| | >12.0 h | 397 (12.4) | 332 (83.9) | 65 (16.1) | | 1.37 (0.90–2.13) | 0.152 |
| | Not Received | 2508 (78.2) | 2250 (89.7) | 258 (10.3) | 0.002 | 0.80 (0.56–1.17) | 0.234 |
| **Time to Non-TBI Surgery, N (%)** | 0.0–12.0 h | 164 (5.1) | 146 (90.3) | 18 (9.7) | | ref | |
| | >12.0 h | 194 (6.0) | 178 (91.8) | 16 (8.2) | | 0.73 (0.36–1.48) | 0.382 |
| | Not Received | 2851 (88.8) | 2524 (88.5) | 327 (11.5) | 0.414 | 1.05 (0.65–1.80) | 0.847 |

A majority of patients waited 0.0 to 1.0 hours after physician evaluation to receive an x-ray (51.3) (Table 2). Most severe GCS patients (61.3%) received an x-ray within 1.0 hours of evaluation whereas most mild (50.4%) and moderate (53.2%) GCS patients received an x-ray at least 1.1 hours after evaluation (Fig 2). We found no significant associations between time to x-ray and in-hospital outcome in our adjusted regression model (Fig 4).

Few patients in our sample received a brain CT scan (4.5%) (Table 2). Most patients did not need and did not receive a CT scan (71.2%). However, a large proportion of patients needed and did not receive a CT scan (24.2%). A majority of severe (95.0%) and moderate (92.1%) GCS patients did not receive a CT scan when needed (Fig 2). We found no significant associations between time to brain CT scan and in-hospital outcome in our adjusted regression model (Fig 4).

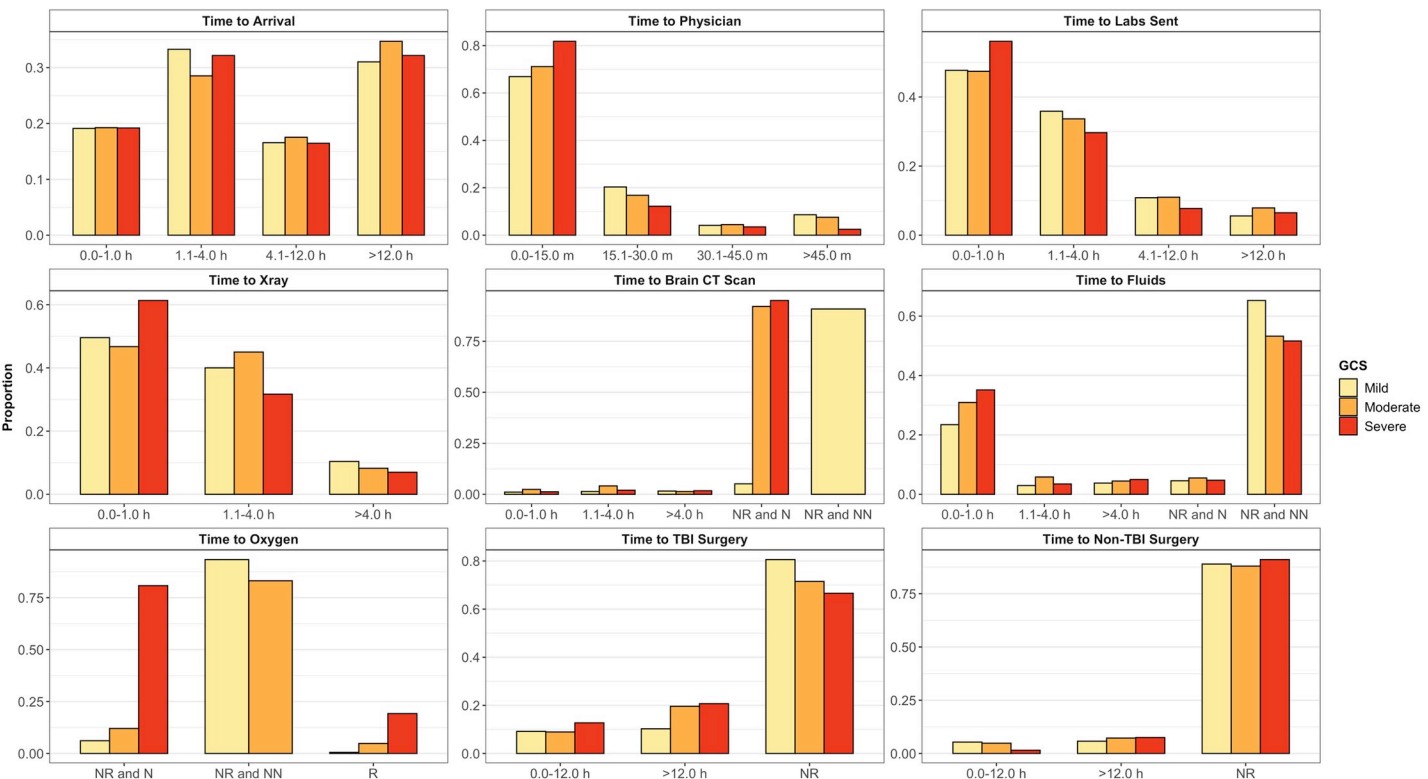

**Fig 2. Proportion of mild, moderate, and severe GCS patients in each level of time to care variables.** R = Received. NR = Not Received. N = Needed. NN = Not Needed.

### Time to treatments

One-fourth of patients in our sample received fluids within 1.0 hours of physician evaluation (25.6%). This includes 23.4% of mild, 30.9% of moderate, and 35.2% of severe GCS patients (Fig 2). However, a majority of patients did not need and did not receive fluids (62.5%) including most mild (65.3%), moderate (53.3%), and severe (51.6%) GCS patients. In our adjusted regression model, not needing and not receiving fluids was protective against a poor outcome in comparison to those who received fluids within 1.0 hours of evaluation (OR: 0.62; 95% CI: 0.44, 0.86) (Fig 4).

Only 3.2% of patients received oxygen (Table 2). Most of these patients had a severe GCS (74.0%) (Fig 2). A majority of patients did not need and did not receive oxygen (80.8%), including 93.4% of mild, 83.2% of moderate, and 0.0% of severe patients. However, 16.0% of patients needed and did not receive oxygen. This includes 6.1% of mild, 12.0% of moderate, and 80.8% of severe GCS patients. In our adjusted regression model, not needing and not receiving oxygen was protective against a poor outcome in comparison to patients who received oxygen (OR: 0.37; 95% CI: 0.18, 0.77) (Fig 4). Needing and not receiving oxygen was also protective against a poor outcome (OR: 0.40; 95% CI: 0.23, 0.70).

Most patients did not receive a TBI (78.2%) or non-TBI (88.8%) surgery (Table 2). TBI surgery recipients included 19.5% of mild, 28.5% of moderate, and 33.4% of severe GCS patients (Fig 2). Non-TBI surgery recipients included 11.1% of mild, 12.0% of moderate, and 9.0% of severe GCS patients. In our adjusted regression model, waiting more than 12.0 hours after physician evaluation for a non-TBI surgery was protective against a poor outcome in comparison to patients who waited 12.0 hours or less (OR: 0.32; 95% CI: 013, 0.79) (Fig 4).

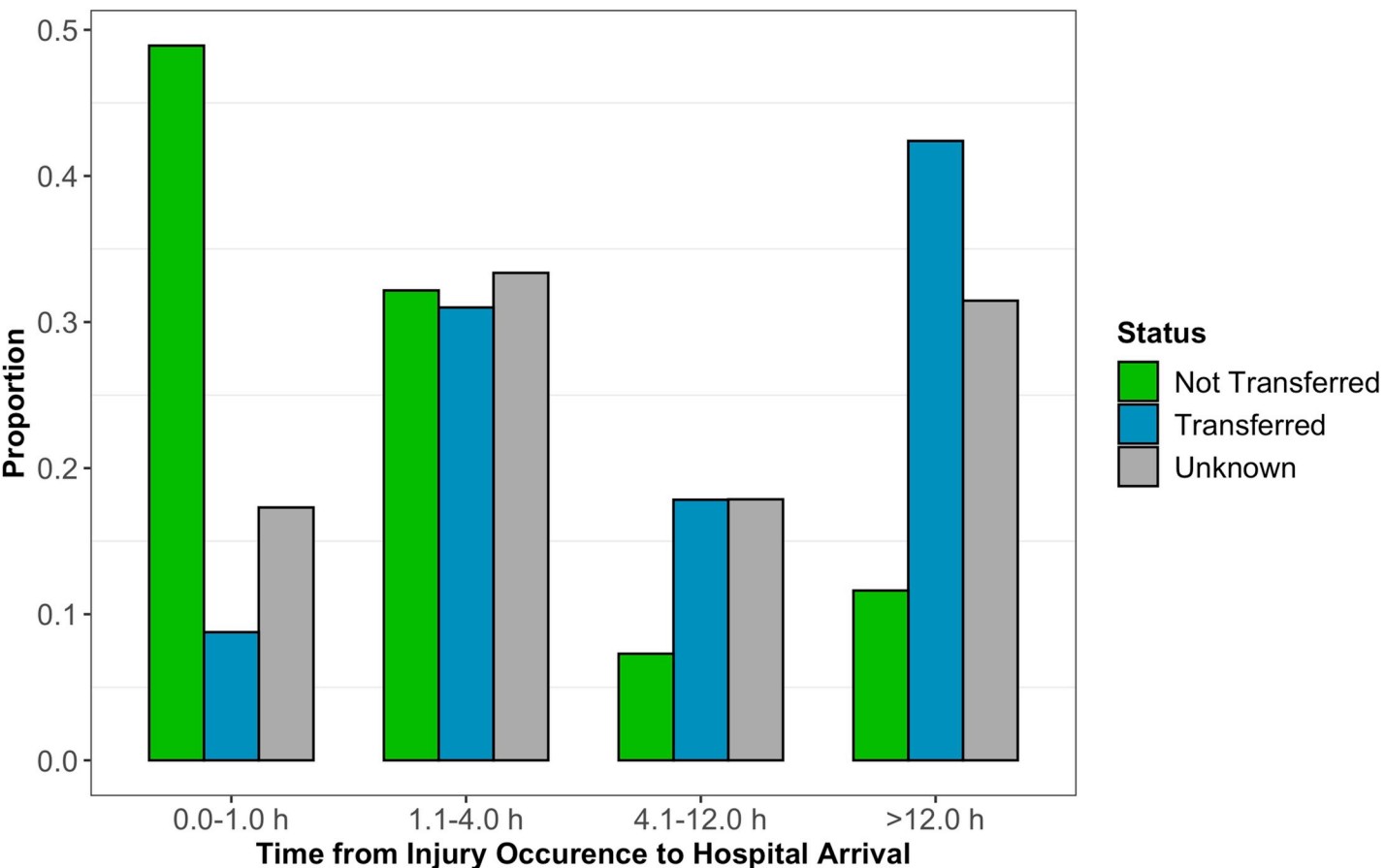

**Fig 3. Proportion of non-transfers and transfers who experienced delays to hospital arrival following injury occurrence.**

## Discussion

To our knowledge, this study is the first to investigate the epidemiology of pre- and in-hospital care delays among trauma patients in sub-Saharan Africa. Understanding the epidemiology of care delays experienced by ED patients is an important step in improving emergency care quality in low-resource hospitals. Our findings from a low-income national referral hospital in Tanzania offer insight into the patterns and consequences of care delays endured by trauma patients in LMICs. First, the status of pre-hospital care and hospital transfer systems in low-income settings may conceal differences between mild, moderate, and severely injured patients with regard to hospital arrival delays and their effect on trauma outcomes. Second, in-hospital delays experienced by trauma patients in low-income settings may be driven exclusively by constraints to non-personnel related resources rather than personnel related resources. Third, the easily accessible nature of time to care data makes it a potential tool for improving real time resource allocation, and thus trauma care, in low-resource hospitals.

### Delays to hospital arrival

An estimated 91.3% of Africa's population lacks access to emergency medical services, including prehospital care [48]. Consequently, individuals who sustain TBI must acquire transport through relatives, friends, bystanders, or commercial drivers [49]. Thus, severely injured patients are expected to experience longer delays to hospital arrival because they have greater

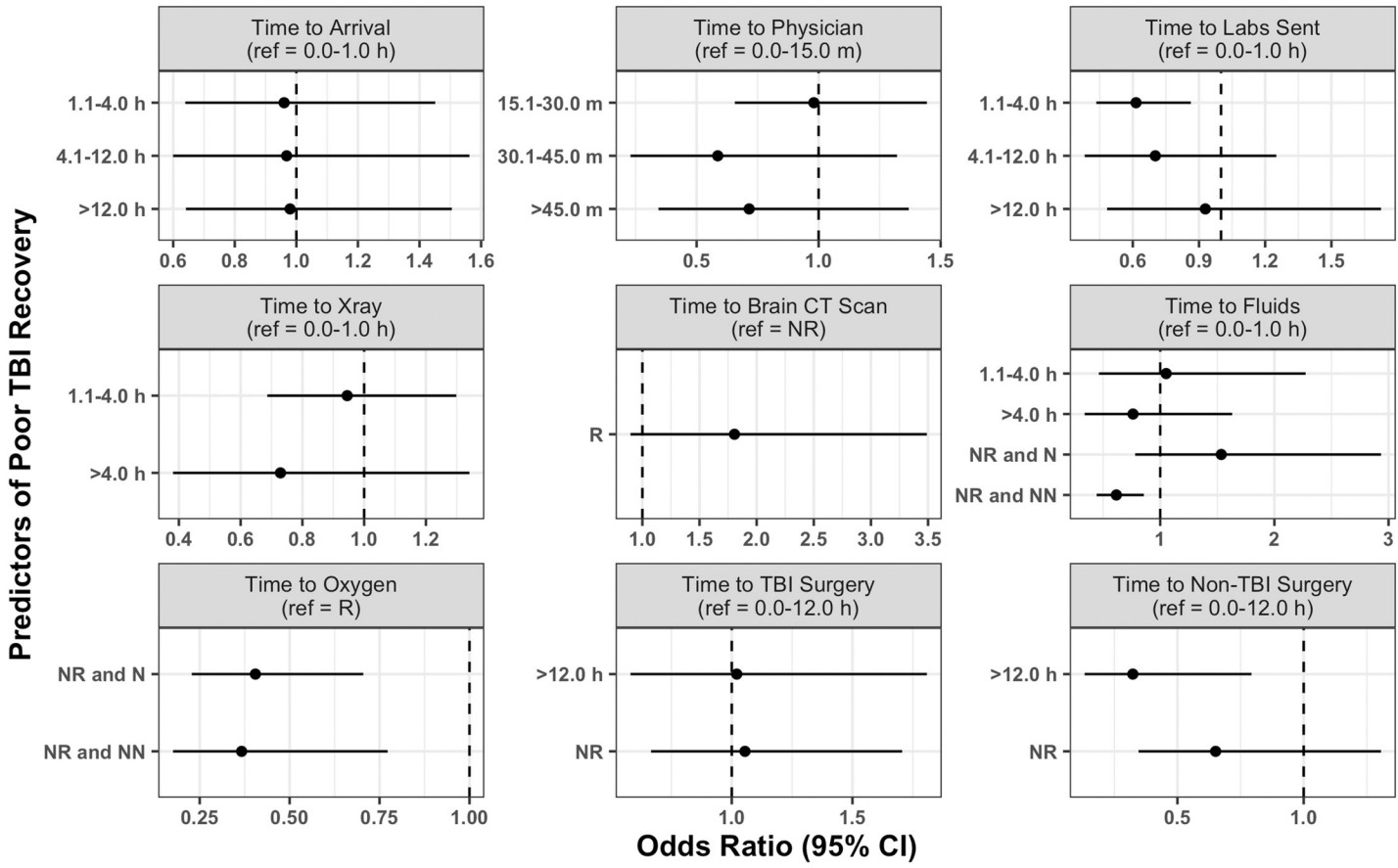

**Fig 4. Results of adjusted logistic regression between time to care variables and in-hospital outcome.** Odds ratios show the odds of a poor outcome. R = Received. NR = Not Received. N = Needed. NN = Not Needed.

difficulty arranging for transport. However, in our study, delays to hospital arrival were not significantly different between mild, moderate, and severe GCS patients (p = 0.768). We have two possible explanations for this result. First, as suggested by the literature, severely injured patients are more likely to succumb to airway compromise, respiratory failure, or uncontrolled hemorrhaging without prompt prehospital care [13]. A significant number of severely injured patients in countries without prehospital services may therefore die before reaching a hospital [50]. Second, a large percentage of patients in our registry were transferred to KCMC from other hospitals (Table 1). A sub-analysis of our data evaluating transfer status and time to hospital arrival revealed that patients with longer arrival delays tend to be transfers (p<0.001). Transferred patients must be able to survive transport and may therefore comprise primarily less severe presentations depending on geographical and contextual factors [51]. Thus, even severely injured patients who do reach their nearest hospital may not appear in our registry as we are a regional referral hospital.

## Access to hospital resources

According to SATS, the ideal time from hospital arrival to physician evaluation for emergency, very urgent, and urgent trauma cases is immediately, less than 10 minutes, and less than 1.0 hours respectively [52]. The evaluation of nearly all severe GCS patients in our sample was within the timeframe recommended by SATS for emergency and very urgent cases; this

demonstrates efficient triage prioritizing the critically injured. A comprehensive evaluation of 364 EDs in the United States revealed an average evaluation wait time of 31.8 minutes for cases classified as emergent (see in 0.0 to 14.0 minutes) [20]. Collectively, these high resource EDs failed to achieve national target wait times for their most time-sensitive patients. Nearly 70.0% of all patients in our sample (including 81.8% of all severely injured patients) were evaluated in 0.0 to 15.0 minutes, suggesting shorter wait times than what has been observed in the United States. Factors affecting patient inflow, notably ED overcrowding and triaging, are therefore unlikely causes of poor in-hospital outcomes among our sample.

Compared to personnel associated resources, our health system was more limited by non-personnel related resources as seen in other LMIC hospitals and which has been associated with poor outcomes [53, 54]. In our sample, 92.0% of moderate and 95.0% of severe GCS patients in our sample needed but did not receive a brain CT scan (Fig 2). Patients must be in a stable condition to undergo transport for medical imaging. Patient stability may explain the disparity we observed between mild and moderate/severe cases. However, in general, patient access to medical imaging is limited across countries in sub-Saharan Africa. In 2012, 100% of Kenya's radiologists were located only in cities and major towns leaving the rural population at a disadvantage [55]. In 2011, only 56.0% of urban patients and 13.0% of rural patients in Uganda who required medical imaging were imaged [56]. In 2014, an evaluation of registered radiological equipment in Tanzania estimated 5.7 general radiography units per one million people; significantly lower than the WHO recommended 20 units per one million people [57]. Similar conditions restricting access to medical imaging may exist in other sub-Saharan African countries. The causes of low medical imaging access may include inequitable access to medical education among rural communities, a lack of radiological training interventions, limited human resources, and low health insurance coverage [55, 58]. Although we found no significant association between CT scan access and in-hospital outcome, the deficiency of CT scans for those most in need among our sample and the documented shortage of diagnostic imaging procedures across other sub-Saharan African countries warrant further investigation into the need for medical imaging equipment in sub-Saharan Africa and its impact on patient outcomes.

Nearly 81.0% of severe GCS patients in our sample needed but did not receive oxygen (Fig 2). Hypoxia has long been known to worsen TBI outcomes [59]. However, patients in our sample who needed and did not receive oxygen had lower odds of experiencing a poor in-hospital outcome (OR: 0.40; 95% CI: 0.23, 0.70). We believe that in the setting of limited monitoring capacity, oxygen may have been administered on the basis of visual rather than diagnostic indications, decreasing the likelihood of oxygen administration to patients with mild or visually undetectable hypoxia. In addition, for those who received oxygen, oxygen saturation may have been recorded into our registry either before or after oxygen administration. Consequently, we are unable to disaggregate non-hypoxic from hypoxic patients among those who received oxygen. As a result of these limitations, our reported associations between oxygen receipt and outcome status may not reflect the true relationship between these two variables. The need to address supplemental oxygen shortages in LMIC hospitals has been well documented [60, 61]. However, the impact of oxygen administration delays on TBI patient outcomes in LMICs has not been established and would benefit from continued investigations using more focused registries.

## Surgical data

Only 21.8% of patients in our sample received a TBI surgery, including 28.5% of moderate and 33.4% of severe GCS patients (Fig 2). Observational studies conducted in neighboring regions

of Tanzania have found even higher proportions of surgery recipients among moderate and severe TBI patients [36, 37]. Similar studies in Rwanda and Uganda have also documented higher rates of surgery among TBI patient cohorts [11, 62]. It is possible that we observed a lower provision of surgery among our sample because patients in our registry did not need surgery. Alternatively, the provision of surgery at KCMC may not be optimized such that those most in need are the ones receiving surgical treatment. Unfortunately, limitations to our data prevent us from determining a patient's need for surgery and therefore which of these two possibilities is more likely. Regardless, the optimization of surgical provision remains an important aspect of TBI care both in Tanzania and in LMICs globally [63–65].

Tanzania has recently implemented reforms to build on its low national neurosurgical capacity [66]. In the meantime, innovative solutions are required to optimize surgical care. Most notably, prognostic models designed to improve triage through the prediction of TBI patient outcomes can assist in allocating surgical care to those who will most likely benefit [67]. The implementation of TBI registries in LMIC hospitals provides an opportunity to harness time to care data with the aim of optimizing the prediction performance of such innovative tools. Although the relationships presented in this study would benefit from further epidemiological investigations, time to care data is easy to collect and should be considered during the development of clinical decision-making technologies. Care delays could prove to be powerful predictors of TBI outcomes.

## Limitations

Interpretation of our study's results is not complete without careful consideration of limitations. First, the TBI registry used for analysis likely underrepresents the most severe injury cases who are more likely to die before reaching a hospital and are therefore less likely to appear in a hospital registry. While care delays may have the greatest impact on this patient population, our registry only captured patients who were well enough to reach the hospital. Since we would expect care delays to have the greatest impact on severely injured patients, we believe that a survivor bias inherent in our study would drive our measures of association towards the null.

Second, our variable time to hospital arrival was based on self-reported or family-reported information regarding the time of injury occurrence. This form of information bias may have led to exposure misclassification with regard to our variable time to hospital arrival. However, because all time information was recorded in the registry prior to ascertainment of patient outcomes, exposure misclassification was independent of patient outcome status. This form of non-differential exposure misclassification would bias our measure of association between time to hospital arrival and outcome status towards the null. Thus, we would expect the true association between hospital arrival delays and patient in-hospital outcome status to be stronger than what is reported in our study.

Finally, our analysis only considered patients presenting to the KCMC ED in Moshi, Tanzania. Given that KCMC is a regional referral hospital and the third largest hospital in the country, the quality of care provided at KCMC is likely preferable to care offered at most other hospitals in the Kilimanjaro region. Care delays may therefore have a larger impact on patients in these other hospitals if care quality is a mediator between care delays and patient outcomes.

## Conclusion

Reducing delays to care is an important component of improving outcomes for all trauma patients. This study provides insight into the epidemiology of care delays among trauma patients presenting to a low-income tertiary referral hospital in sub-Saharan Africa. There is a

paucity of literature on the subject of care delays in LMICs. Future studies should continue investigating the association between care delays and trauma patient outcomes in low-resource settings. Time to care data is easy to collect as it requires no tools and only two data points for a given patient: time of event A and time of event B. Time to care data collection may therefore be effortlessly integrated into hospital registries in any context.

## Author Contributions

**Conceptualization:** Blandina T. Mmbaga, Joao Ricardo Nickenig Vissoci, Catherine A. Staton.

**Data curation:** Mark Mvungi, Blandina T. Mmbaga, Francis Sakita.

**Formal analysis:** Armand Zimmerman, Randi Griffin, Erika Bárbara Abreu Fonseca Thomaz.

**Funding acquisition:** Charles J. Gerardo.

**Methodology:** Armand Zimmerman.

**Supervision:** Blandina T. Mmbaga, Francis Sakita, Charles J. Gerardo, Joao Ricardo Nickenig Vissoci, Catherine A. Staton.

**Visualization:** Armand Zimmerman, Erika Bárbara Abreu Fonseca Thomaz.

**Writing – original draft:** Armand Zimmerman, Samara Fox.

**Writing – review & editing:** Armand Zimmerman, Taylor Nelp, Blandina T. Mmbaga, Francis Sakita, Charles J. Gerardo, Joao Ricardo Nickenig Vissoci, Catherine A. Staton.

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
