## [Decision Letter · Decision Letter 0]

25 Jun 2020

PONE-D-20-14993

An Analysis of Emergency Care Delays Experienced by Traumatic Brain Injury Patients Presenting to a Regional Referral Hospital in a Low-Income Country

PLOS ONE

Dear Dr. Staton,

Thank you for submitting your manuscript to PLOS ONE. After careful consideration, we feel that it has merit but does not fully meet PLOS ONE’s publication criteria as it currently stands. Therefore, we invite you to submit a revised version of the manuscript that addresses the points raised during the review process.

Please, clarify the points addressed b reviewer 1 to improve your manuscript.

We look forward to receiving your revised manuscript.

Kind regards,

Hans-Peter Simmen, M.D., Professor of Surgery

Academic Editor

PLOS ONE

Journal Requirements:

2. Please provide additional details regarding participant consent.

In the ethics statement in the Methods and online submission information, please ensure that you have specified (i) whether consent was informed and (ii) what type you obtained (for instance, written or verbal, and if verbal, how it was documented and witnessed).

If your study included minors, state whether you obtained consent from parents or guardians.

If the need for consent was waived by the ethics committee, please include this information.

Reviewers' comments:

Reviewer's Responses to Questions

**Comments to the Author**

1. Is the manuscript technically sound, and do the data support the conclusions?

Reviewer #1: Partly

Reviewer #2: Yes

2. Has the statistical analysis been performed appropriately and rigorously? 

Reviewer #1: Yes

Reviewer #2: Yes

3. Have the authors made all data underlying the findings in their manuscript fully available?

Reviewer #1: Yes

Reviewer #2: Yes

4. Is the manuscript presented in an intelligible fashion and written in standard English?

Reviewer #1: Yes

Reviewer #2: Yes

5. Review Comments to the Author

Reviewer #1: Dear authors,

I have read your work with interest since the topic is of enormous importance. I think the manuscript is well-written. The compact title reflects the manuscript well. The introduction is clear and with sufficient references.

While reading your draft the following questions came to my mind:

A) Most importantly I would like you to clarify the timepoint at which you measured the outcome of your patients. How long was the time from injury to measuring the GOS on average with SD? The time to follow-up is not mentioned at all! Please adapt this aspects of your draft accordingly!

B) Regarding the reported results (line 197 ff): Please do not only differentiate between good and poor results regarding the GOS but also report on the mortality rates.

C) In Table 1 and Figure 3 you mention the possibility of an unknown Transfer Status-> please clarify.

D) In line 284 to 289 you state "Most patients did not receive a TBI (78.2%) or non-TBI (88.8%) surgery (Table 2). TBI surgery recipients included 19.5% of mild, 28.5% of moderate, and 33.4% of severe GCS patients (Figure 2). Non-TBI surgery recipients included 11.1% of mild, 12.0% of moderate, and 9.0% of severe GCS patients. In our adjusted regression model, waiting more than 12.0 hours after physician evaluation for a non-TBI surgery was protective against a poor outcome in comparison to patients who waited 12.0 hours or less (OR: 0.32; 95% CI: 013, 0.79)". What kind of TBI surgeries where performed based on what kind of decision making process? What did the injury pattern of the non-TBI surgery recipients look like and what kind of surgical interventions was performed?

E) In line 365 and 366 you state "However, for those who received oxygen,

366 oxygen saturation may have been recorded into our registry either before or after administration." If you are not able to say whether the oxygen saturation was measured before or after administration, how reliable is this information overall?

Reviewer #2: The authors report a retrospective analysis of acute TBI patients from a prospective patient registry of 3209 cases from the time period 2013 - 2017. The report is well written, the analyses are sound and the reported findings are interesting, original and important.

The major finding is, that improvement of TBI outcome (of those patients who live to see a physician) requires improvement of non-personnel related resources (availability of neuroimaging, oxygen and surgery).

6. PLOS authors have the option to publish the peer review history of their article (what does this mean?). If published, this will include your full peer review and any attached files.

Reviewer #1: No

Reviewer #2: **Yes: **Lennart Stieglitz

---

## [Author Response · Author response to Decision Letter 0]

18 Sep 2020

July 20, 2020

To the editors and reviewers of PLOS ONE,

Thank you for the opportunity to edit and revise our manuscript. We have tried to address each reviewer comment below.

Reviewer #1

Reviewer Comment: Dear authors, I have read your work with interest since the topic is of enormous importance. I think the manuscript is well-written. The compact title reflects the manuscript well. The introduction is clear and with sufficient references. While reading your draft the following questions came to my mind:

Reply: Thank you very much for this comment.

Reviewer Comment: A) Most importantly I would like you to clarify the timepoint at which you measured the outcome of your patients. How long was the time from injury to measuring the GOS on average with SD? The time to follow-up is not mentioned at all! Please adapt this aspects of your draft accordingly!

Reply: Thank you for this excellent point. GOS was measured at the time of hospital discharge. The mean time from injury to hospital discharge was 4.8 days (SD: 4.6 days). The mean time from injury to death was 3.6 days (SD: 3.5 days). This information has been included in the methods and results section of our revised manuscript.

Our dataset does not capture follow-up information as our TBI registry was part of a study designed to define areas for quality improvement in the initial management of TBI patients at KCMC. Our last point of contact with patients included in the registry was hospital discharge. This was clarified in the methods section of our revised manuscript.

Reviewer Comment: B) Regarding the reported results (line 197 ff): Please do not only differentiate between good and poor results regarding the GOS but also report on the mortality rates.

Reply: Thank you for this comment. Among the 361 patients who experienced a poor outcome, 326 patients (90.3%) died. Consequently, results differentiating between those who survived and those who died are not significantly different from results differentiating between those who experienced a good outcome and those who experienced a poor outcome. We therefore chose only to differentiate between good and poor outcomes. We have included in the results section of our revised manuscript a sentence stating the number of deaths in the poor outcome group as well as our reason for reporting results as such in the methods section of our revised manuscript.

Reviewer Comment: C) In Table 1 and Figure 3 you mention the possibility of an unknown Transfer Status-> please clarify.

Reply: Thank you for this comment. Our variable Transfer Status was categorized as transferred from a previous hospital or not transferred from a previous hospital. Patients with missing prehospital course information were categorized as having an unknown transfer status, as we could not determine whether or not these patients were transferred to KCMC from a previous hospital. An explanation of the unknown transfer status category has been included in the methods section of our revised manuscript.

Reviewer Comment: D) In line 284 to 289 you state "Most patients did not receive a TBI (78.2%) or non-TBI (88.8%) surgery (Table 2). TBI surgery recipients included 19.5% of mild, 28.5% of moderate, and 33.4% of severe GCS patients (Figure 2). Non-TBI surgery recipients included 11.1% of mild, 12.0% of moderate, and 9.0% of severe GCS patients. In our adjusted regression model, waiting more than 12.0 hours after physician evaluation for a non-TBI surgery was protective against a poor outcome in comparison to patients who waited 12.0 hours or less (OR: 0.32; 95% CI: 013, 0.79)". What kind of TBI surgeries where performed based on what kind of decision making process? What did the injury pattern of the non-TBI surgery recipients look like and what kind of surgical interventions was performed?

Reply: Thank you for this comment. TBI surgeries are limited to burr holes, craniotomies, and craniectomies conducted by trained general surgeons as there are no neurosurgeons stationed at KCMC. However, our registry does not include information on the specific type of TBI surgery received by a patient. The decision to operate on a TBI patient and what kind of operation to perform is ultimately based on the judgement of a general surgeon trained in neurosurgical procedures. Alternatively, a patient or patient’s family may choose to forgo surgery if the cost cannot be covered. Again, our registry does not include information on the decision process used to inform the provision of surgery for a given patient. This information has been included in the methods section of the revised manuscript.

With regard to non-TBI surgeries, our registry does not include the types of operations performed on non-TBI surgery recipients nor does it include the injury patterns of non-TBI surgery recipients. Based on other injury registries currently established at KCMC, we do know that non-TBI surgeries include primarily otolaryngology, general, and orthopedic operations. However, we do not have specific details of these procedures and these procedures may not represent the operations received by non-TBI surgery recipients in our dataset. Thus, we chose not to include this information in our manuscript.

Reviewer Comment: E) In line 365 and 366 you state "However, for those who received oxygen, oxygen saturation may have been recorded into our registry either before or after administration." If you are not able to say whether the oxygen saturation was measured before or after administration, how reliable is this information overall?

Reply: Thank you for this comment. Among patients who received oxygen, we cannot be certain whether oxygen saturation was measured before or after oxygen administration. Although this limitation makes our oxygen saturation data non-optimal for these patients, it has the effect of underestimating potential hypoxia, and it may help explain why we observed an unintuitive association between oxygen receipt and a worse outcome status. For example, a patient with a poor outcome may have presented with hypoxia, received oxygen, and then had their oxygen saturation measured. Thus, this hypoxic patient would appear in our registry as non-hypoxic. We have included a statement in the discussion that addresses the reliability of our results regarding our reported associations between oxygen receipt and outcome status.

Reviewer #2

Reviewer Comment: The authors report a retrospective analysis of acute TBI patients from a prospective patient registry of 3209 cases from the time period 2013 - 2017. The report is well written, the analyses are sound and the reported findings are interesting, original and important. The major finding is, that improvement of TBI outcome (of those patients who live to see a physician) requires improvement of non-personnel related resources (availability of neuroimaging, oxygen and surgery).

Reply: Thank you very much for this comment.

---

## [Editor Report · Decision Letter 1]

29 Sep 2020

An analysis of emergency care delays experienced by traumatic brain injury patients presenting to a regional referral hospital in a low-income country

PONE-D-20-14993R1

Dear Dr. Staton,

We’re pleased to inform you that your manuscript has been judged scientifically suitable for publication and will be formally accepted for publication once it meets all outstanding technical requirements.

Kind regards,

Hans-Peter Simmen, M.D., Professor of Surgery

Academic Editor

PLOS ONE
---

## [Editor Report · Acceptance letter]

2 Oct 2020

PONE-D-20-14993R1 

An analysis of emergency care delays experienced by traumatic brain injury patients presenting to a regional referral hospital in a low-income country 

Dear Dr. Staton:

I'm pleased to inform you that your manuscript has been deemed suitable for publication in PLOS ONE. Congratulations! Your manuscript is now with our production department. 

Kind regards, 

on behalf of

Dr. Hans-Peter Simmen 

Academic Editor

PLOS ONE